# Type 3 Inositol 1,4,5-Trisphosphate Receptor is a Crucial Regulator of Calcium Dynamics Mediated by Endoplasmic Reticulum in HEK Cells

**DOI:** 10.3390/cells9020275

**Published:** 2020-01-22

**Authors:** Lili Yue, Liuqing Wang, Yangchun Du, Wei Zhang, Kozo Hamada, Yoshifumi Matsumoto, Xi Jin, Yandong Zhou, Katsuhiko Mikoshiba, Donald L. Gill, Shengcheng Han, Youjun Wang

**Affiliations:** 1Beijing Key Laboratory of Gene Resource and Molecular Development, College of Life Sciences, Beijing Normal University, Beijing 100875, China; 201831200005@mail.bnu.edu.cn (L.Y.); 201931200006@mail.bnu.edu.cn (L.W.); 201831200004@mail.bnu.edu.cn (Y.D.); jinxi@bnu.edu.cn (X.J.); 2Department of Molecular and Cellular Biology, College of Biological Science, University of Guelph, 50 Stone Rd E, Guelph, ON N1G 2W1, Canada; weizhang@uoguelph.ca; 3Shanghai Institute for Advanced Immunochemical Studies, ShanghaiTech University, 393 Middle Huaxia Road, Shanghai 201210, China; hamada@shanghaitech.edu.cn (K.H.); mikosiba@shanghaitech.edu.cn (K.M.); 4Central Institute for Experimental Animals, Kawasaki, Kanagawa 210-0821, Japan; miraihe08@brain.riken.jp; 5Department of Cellular and Molecular Physiology, Pennsylvania State University College of Medicine, Hershey, PA 17033, USA; yzhou1@pennstatehealth.psu.edu (Y.Z.); dgill4@pennstatehealth.psu.edu (D.L.G.)

**Keywords:** IP_3_R, store-operated Ca^2+^ entry, Orai1, SERCA, NEDD4L, calcium

## Abstract

Being the largest the Ca^2+^ store in mammalian cells, endoplasmic reticulum (ER)-mediated Ca^2+^ signalling often involves both Ca^2+^ release via inositol 1, 4, 5-trisphosphate receptors (IP_3_R) and store operated Ca^2+^ entries (SOCE) through Ca^2+^ release activated Ca^2+^ (CRAC) channels on plasma membrane (PM). IP_3_Rs are functionally coupled with CRAC channels and other Ca^2+^ handling proteins. However, it still remains less well defined as to whether IP_3_Rs could regulate ER-mediated Ca^2+^ signals independent of their Ca^2+^ releasing ability. To address this, we generated IP_3_Rs triple and double knockout human embryonic kidney (HEK) cell lines (IP_3_Rs-TKO, IP_3_Rs-DKO), and systemically examined ER Ca^2+^ dynamics and CRAC channel activity in these cells. The results showed that the rate of ER Ca^2+^ leakage and refilling, as well as SOCE were all significantly reduced in IP_3_Rs-TKO cells. And these TKO effects could be rescued by over-expression of IP_3_R3. Further, results showed that the diminished SOCE was caused by NEDD4L-mediated ubiquitination of Orai1 protein. Together, our findings indicate that IP_3_R3 is one crucial player in coordinating ER-mediated Ca^2+^ signalling.

## 1. Introduction

Calcium ion (Ca^2+^) is essential for life, regulating many crucial cellular functions like contraction, migration and proliferation [1]. Cytosolic Ca^2+^ signals are mainly controlled through a combination of two closely linked processes–Ca^2+^ release from the endoplasmic reticulum (ER) and Ca^2+^ entry across the plasma membrane (PM) [2]. Much of the former event is mediated by inositol 1,4,5-trisphosphate receptor (IP_3_Rs), Ca^2+^ release channels resident in ER membranes [3,4,5,6]. As a major Ca^2+^ release channel on ER, the largest intracellular Ca^2+^ store in mammalian cells [7], IP_3_Rs often form large complexes with other Ca^2+^ handling proteins and Ca^2+^ channels [8]. Numerous reports have showed biochemical or co-localization evidence indicating that IP_3_Rs interact with PM Ca^2+^-transporting ATPase (PMCA) [9,10], Sarco/endoplasmic reticulum Ca^2+^ ATPase 2 (SERCA2) or SERCA3 [10,11], STIM1 [12], Orai1 channels [13,14], and handling organelles like mitochondria [15] in different types of cells. Thus, IP_3_Rs are key elements of Ca^2+^ signalling machinery. IP_3_Rs interact with these proteins to spatiotemporally control Ca^2+^ signals and corresponding downstream events [8], aberrant IP_3_R-regulated-signalling are related to many types of human diseases [16]. Yet it still remains elusive how IP_3_Rs would coordinate these Ca^2+^ handling proteins to maintain ER Ca^2+^ homeostasis.

In mammalian cells, one ubiquitous Ca^2+^ entry pathway mediated by Ca^2+^ release activated Ca^2+^ (CRAC) channels is also essential for the maintaining of ER Ca^2+^ homeostasis [17,18]. Its activation is closely linked with the function of IP_3_Rs [19,20]. Activation of G-protein coupled receptors or Tyrosine kinase on PM would lead to the opening of immobile IP_3_Rs adjacent to ER-PM junctions [21], resulting in a transient increase in cytosolic Ca^2+^ levels and a decrease in ER Ca^2+^ content. The lowering of ER Ca^2+^ store would induce Ca^2+^ influxes from extracellular space through a process called store operated Ca^2+^ entry (SOCE), resulting in a sustained increase cytosolic Ca^2+^ levels [2]. SOCE is mediated by CRAC channels. Briefly, a decrease in ER Ca^2+^ levels is sensed by STIM1, the ER-resident Ca^2+^ sensing component of the CRAC channel. Activated STIM1 will then translocate into ER-PM junctions to bind with and activate Orai1, the pore forming subunit of CRAC channels, causing Ca^2+^ influxes [18]. Thus the activation of IP_3_Rs and CRAC channels is spatiotemporally coupled around ER-PM junctions [22]. Even though Ca^2+^ release signals generated by IP_3_Rs and Ca^2+^ influxes through CRAC channels occur at different subcellular regions, mediating different cellular processes [23,24], IP_3_Rs and CRAC channels are functionally coupled [24,25,26]. There is one report showing that mutation in IP_3_Rs could inhibit SOCE in *Drosophila* cells, likely through impairments of coupling between STIM1-Orai1 [27]. However, it still remains elusive whether IP_3_Rs could regulate SOCE through other means.

To further dissect IP_3_Rs-centered, ER-mediated Ca^2+^ signalling in a genetic clean background, we generated IP_3_Rs triple and double knockout HEK cell lines (IP_3_Rs-TKO and IP_3_Rs-DKO) using CRISPR/Cas9 genome-editing technology. Using these engineered cells together with ER Ca^2+^ indicator CEPIA1ers (Calcium-measuring organelle-Entrapped Protein IndicAtor 1 in the ER) [28], we demonstrated that even though IP_3_Rs-TKO cells managed to keep normal ER Ca^2+^ homeostasis, they had impaired ER-Ca^2+^ dynamics and diminished SOCE. Our results showed that the expression of IP_3_R3 correlated with the rate of ER Ca^2+^ leakage and refilling, and that IP_3_R3 affected SOCE by regulating NEDD4L (neural precursor cell expressed developmentally downregulated gene 4-like)-mediated ubiquitination of Orai1 protein. Overall, our findings suggest that IP_3_R3 maybe one key player in coordinating ER-mediated Ca^2+^ signalling.

## 2. Results and Discussion

### 2.1. With IP_3_R2 Being the Dominant Isoform, IP_3_Rs Regulate Growth and Migration of HEK Cells

To examine the role of IP_3_Rs in Ca^2+^ signalling, we made two different IP_3_R1-2-3 triple knockout HEK cell lines (IP_3_Rs-TKO) with CRISPR/Cas9 genomic editing technology using procedures similar to previous reports [17]. These IP_3_Rs-TKO cells were generated from two separate HEK cell lines stably expressing genetically encoded Ca^2+^ indicators (GECI): GCaMP6m [29], a cytosolic Ca^2+^ indicator; or R-CEPIA1er, an ER Ca^2+^ indicator [28]. Thus, they were named as GIPK (GCaMP6m cells with IP_3_Rs-TKO) or RIPK (R-CEPIA1er cells with IP_3_Rs-TKO). After confirming the effectiveness of knock-out with sequencing (Appendix A), we tested their responses to 100 μM carbachol (CCh), an agonist for muscarinic acetylcholine receptor that could lead to IP_3_ release. In R-CEPIA1er cells transiently expressing GCaMP6m, CCh could induce Ca^2+^ release from ER, as indicated by a decrease in R-CEPIA1er signal (ER Ca^2+^ levels, red trace) and a transient increase in G-CaMP6m signal (cytosolic Ca^2+^ levels, green trace) (Figure 1A, left panel). However, these two events were completely abolished in RIPK cells (Figure 1A, right panel). Similarly, CCh also failed to increase cytosolic Ca^2+^ levels in GIPK cells, as indicated by no CCh-induced increase in GCaMP fluorescence (Figure 1B). Together, these results reveal that all three IP_3_Rs were functionally knocked-out in both RIPK and GIPK cells.

To characterize the relative contribution of each IP_3_R isoform in IP_3_-induced Ca^2+^ release in HEK cells, we also generated double knockout cell lines (IP_3_Rs-DKO) using GCaMP6m cells (Appendix A). The results showed that, similar to an earlier report [30], cells with endogenous IP_3_R2 could mediate CCh-induced Ca^2+^ release with amplitudes similar to those of control cells, while those bearing endogenous IP_3_R1 or IP_3_R3 had much smaller capability in releasing Ca^2+^ (71% and 15% relative to control, correspondingly) (Figure 1C). To estimate Ca^2+^ release more accurately, 1-mM GdCl_3_ were used to block all Ca^2+^ transports across PM [31]. We then checked whether overexpressed of IP_3_Rs, especially IP_3_R1, could fully rescue CCh-induced Ca^2+^ releases in TKO cells. We used P2A system to co-express IP_3_Rs with different fluorescence proteins. These IP_3_R constructs expressed well as indicated by the presence of corresponding fluorescence. GIPK or RIPK cells transiently overexpressing any IP_3_R isoform could fully rescue the CCh-induced Ca^2+^ release (Figure 1D and Appendix A). As indicated by increases in GEM-GECO signals, RIPK cells with overexpressed IP_3_R1 had even larger CCh-induced Ca^2+^ releases than WT cells (Figure 1D and Appendix A), likely caused by higher protein levels of overexpressed-IP_3_R1. Together with one previous report showed that all three IP_3_Rs could generate Ca^2+^ puffs, the elementary Ca^2+^ release event, similarly [30], these results indicated that the smaller Ca^2+^ responses in DKO cells only expressing IP_3_R1/3 was caused by lower expression levels of corresponding IP_3_R isoforms. Indeed, quantitative RT-PCR results showed that mRNA levels of IP_3_R2 and IP_3_R3 is far higher than that of IP_3_R1 in WT cells. Moreover, the mRNA level of IP_3_R1 in IP_3_R2&3 DKO cells was further reduced by ~40% (Figure 1E). Overall, these results indicate that HEK cells might predominantly express IP_3_R2, and IP_3_R1 only contribute marginally to CCh-induced Ca^2+^ release in HEK cells.

IP_3_Rs-mediated Ca^2+^ signalling is involved in diverse physiological processes, such as fertilization, metabolism, secretion apoptosis, and cell growth [32]. We thus examined the proliferation and migration rate of RIPK cells. Compared to WT cells, the proliferation rate of RIPK cells was significantly reduced (Left panel, Figure 1F) and the migration rate was also minimally affected (Right panel Figure 1F). Similar results were also obtained with GIPK cells (data not shown). These results thus showed that IP_3_Rs are involved in the regulation of growth and migration of HEK cells.

### 2.2. Overexpression of IP_3_R1 or IP_3_R3 Could Rescue the Impaired ER Ca^2+^ Leak and Refilling in IP_3_R-TKO Cells

To examine whether these changes in cell behavior were induced by alterations of ER Ca^2+^ homeostasis in IP_3_Rs-TKO cells, we first estimated the size of ER Ca^2+^ store in RIPK cells by comparing their total cytosolic Ca^2+^ releases from ER with those of R-CEPIA1er cells transiently expressing GEM-GECO1, a ratiometric cytosolic GECI [33]. To avoid artifacts caused by possible altered rate of Ca^2+^ extrusion via PM Ca^2+^ pumps or sodium/Ca^2+^ exchangers, 1 mM GdCl_3_ were included in extracellular solution to block all Ca^2+^ transports across PM [31]. The results showed that there is no detectible difference in thapsigargin (TG, 1 μM)-induced Ca^2+^ releases between WT and RIPK cells (Figure 2A). Fura-2 measurements also failed to detect differences in TG-induced responses between these two types of cells (Appendix A), indicating no changes in ER Ca^2+^ contents in RIPK cells. Consistent with these results, when directly measured with a ratiometric ER Ca^2+^ indicator GEM-CEPIA1er [28], RIPK cells also had similar GEM-CEPIA1er signals to those from R-CEPIA1er cells (Appendix A). All these results showed that RIPK cells successfully maintain a normal ER Ca^2+^ homeostasis. Therefore, even though IP_3_Rs are major Ca^2+^ release channels on ER and were shown required for maintaining cellular energy homeostasis [34], they are not essential for maintaining ER Ca^2+^ homeostasis.

We then checked whether ER Ca^2+^ dynamics were affected by IP_3_Rs-TKO. We first examined the Ca^2+^ leak rate after blocking SERCA activity with 1 μM TG in cells expressing GEM-CEPIA1er. RIPK cells showed significantly slower decrease in GEM-CEPIA1er ratio (Figure 2B), indicating a slower decay rate of ER Ca^2+^ levels. Similarly, the speed of decrease in R-CEPIA1er signals during passive ER store depletion induced by 10 mM EGTA in digitonin permeabilized RIPK cells were also significantly slower than R-CEPIA1er cells (Figure 2C, left panel). Together, these results showed that RIPK cells had a slower ER Ca^2+^ leakage. Since ER Ca^2+^ levels are determined by the relative balance between Ca^2+^ leak by leak channels and refilling by SERCA, un-altered resting ER Ca^2+^ content together with a weaker ER Ca^2+^ leakage would imply a less SERCA pumping activity. We thus directly measured refilling of ER Ca^2+^ store with R-CEPIA1er and a protocol adapted from one previous report [35]. In permeabilized, ER Ca^2+^ store depleted RIPK cells, addition of 1 mM ATP and 100 nM Ca^2+^ would enable the refilling of Ca^2+^ into ER by SERCA, as indicated by increases in R-CEPIA1er signals (Figure 2C). Compared to those of WT cells, the rate of increases in R-CEPIA1er signals in RIPK cells was slower, and the amplitude of increase is significantly smaller (Figure 2D). This result demonstrated that the refilling of ER store is also slower in RIPK cells. Other factors may also affect ER Ca^2+^ dynamics directly or indirectly [36]. For example, Ca^2+^ buffering protein like calreticulin was shown to modulate IP_3_Rs-mediated Ca^2+^ releases and inhibit or delay corresponding SOCE [37,38,39,40]. Further researches are needed to elucidate the effects of IP_3_Rs on them. Nevertheless, these results showed that ER Ca^2+^ dynamics, both leakage and refilling, is slower in IP_3_Rs-TKO cells.

In order to dissect out which isoform is responsible for the modulation of leakage and refilling of ER Ca^2+^ ions, we transiently transfected different IP_3_R isoforms and further examined the refilling and leak rate in permeabilized RIPK or R-CEPIA1er cells. Briefly, we used the resulting maximal increase in R-CEPIA1er signals after the addition of ATP and Ca^2+^ to demonstrate the efficacy of ER Ca^2+^ refilling via SERCA (Figure 2E). After the ER Ca^2+^ levels reached a steady state, we then added TG to block SERCA, and used the rate of decreases R-CEPIA1er signals to indicate the release of Ca^2+^ through leak channels on ER (Figure 2E). Compared to blank control, the maximal changes in amplitude of R-CEPIA1er signals was significantly larger in RIPK cells overexpressing IP_3_R3 (Figure 2F, left panel), showing that the presence of IP_3_R3 may promote ER Ca^2+^ refilling. Consistent with one previous report showing functional couplings between IP_3_R1 and SERCA [41], overexpression of IP_3_R1 also significantly enhanced ER Ca^2+^ refilling (Appendix A, left and middle panels). While overexpression of IP_3_R2 had no such effect (Appendix A, left and middle panels). Together, these results indicate that, besides IP_3_R1, IP_3_R3 may also functionally couple with SERCA to regulate ER Ca^2+^ dynamics. Normally, there is a balance between SERCA activity and ER Ca^2+^ leakage [24], changes in SERCA behavior often lead to corresponding alterations in Ca^2+^ leakage. Indeed, IP_3_R1 and IP_3_R3 could significantly increase the decay rate of TG-induced decreases in R-CEPIA1er signals (Figure 2F, right panel; Appendix A, right panel), while IP_3_R2 had no such effect (Appendix A, right panel). Thus, IP_3_R1 and IP_3_R3 might also functionally couple with ER Ca^2+^ leak channels. Together, these results imply that IP_3_R1 and IP_3_R3 might be crucial players in ER Ca^2+^ signalling machinery, balancing the function of SERCA and Ca^2+^ leak channels on ER. We do not know how the expression of IP_3_Rs would affect the function of SERCA and Ca^2+^ leak channels. IP_3_Rs may exert their effects via direct physical interactions, or through IP_3_Rs-dependent transcriptional changes, or by some post-translational modifications. Further mechanistic studies are needed to get a better understanding on this.

We also examined the effect of IP_3_Rs-TKO on SOCE-mediated refilling of ER Ca^2+^ stores in intact cells. Cells were first bathed in Ca^2+^ free imaging solution containing 300 μM EGTA and 2.5 μM ionomycin for 8 min to get their ER Ca^2+^ stores depleted. Then ionomycin was washed out and the cells were kept in 0Ca^2+^ solution for 10 minutes. Ca^2+^ was then added into extracellular solutions to allow SOCE and corresponding refilling of Ca^2+^ store. Consistent with previous results showing that RIPK cells have normal basal ER Ca^2+^ levels (Figure 2B and Appendix A), SOCE responses of RIPK cells were still large enough to generate normal refilling when supplied with regular concentration of external Ca^2+^ (1 mM) (data not shown). However, when low external Ca^2+^ (300 μM) were applied, the increase rate of R-CEPIA1er signals was dramatically slower in RIPK cells than those of R-CEIPIA1er cells (Figure 2G). Compared with results from permeabilized cells, SOCE-mediated ER Ca^2+^ refilling in intact RIPK cells were further slowed down (Right panel of Figure 2G vs Figure 2D), indicating a smaller SOCE in RIPK cells.

### 2.3. Expression of IP_3_R3 Restored the Reduced Orai1 Protein Level and SOCE in IP_3_Rs-TKO Cells

We thus examined the effect of IP_3_Rs-KO on SOCE in HEK cells. Compared to corresponding WT cells, TG-induced increases in Fura-2 ratio or GCaMP signals were significantly reduced in GIPK and RIPK cells, correspondingly (Figure 3A and Appendix A). These results showed that IP_3_Rs-KO reduced SOCE in HEK cells. This result is consistent with one previous finding obtained with IP_3_R mutants in *Drosophila* neurons [27], but different to an earlier finding in chicken B cells [42]. Thus, these observed correlation between expression of IP_3_Rs and SOCE amplitudes maybe specific to cell types. Nevertheless, IP_3_Rs-DKO cells expressing only endogenous IP_3_R1 showed ~50% reduction in SOCE compared to WT cells. While SOCE in another two DKO cells with endogenous IP_3_R2 or IP_3_R3 expressed was unaltered, indicating their critical roles in regulating SOCE (Appendix A). This result also indicates IP_3_R2 and IP_3_R3 had some redundant roles in SOCE regulation.

We next examined the effect of overexpressing different IP_3_R isoforms on SOCE responses of GIPK or RIPK cells. Consistent with results from DKO cells (Appendix A) and an earlier report [27], overexpression of IP_3_R1 had no effect on SOCE in RIPK cells (Appendix A). Even though GCaMP cells with endogenous IP_3_R2 had normal SOCE, overexpression of IP_3_R2 also failed to rescue SOCE in GIPK cells (Appendix A). This discrepancy is likely caused by some long-term adaptation in DKO cells. Nevertheless, only overexpression of IP_3_R3 could significantly increase SOCE in IP_3_Rs-TKO cells (Figure 3B). Together, these results showed that the expression of IP_3_R3 also could regulate SOCE in HEK cells. It is known that SOCE-mediating CRAC channels, SERCA and IP_3_Rs are functionally coupled under physiological conditions [24,25,26,43]. Our results thus added a new aspect of couplings among Ca^2+^ handling machineries.

We then set out to dissect how IP_3_Rs-TKO would affect SOCE. Inspired by a report showing that overexpression of STIM and Orai restored SOCE in *Drosophila* neurons with IP_3_R mutation [44], we compared the protein levels of STIM1 and Orai1in IP_3_Rs-TKO cells with those of WT cells. Western blotting results showed that the expression of STIM1 was similar, while the protein level of Orai1 was significantly lower in both IP_3_Rs-TKO cell lines (Appendix A). Further results showed that overexpression of IP_3_R3 could significantly increase Orai1 protein levels in GIPK cells (Figure 3C). These findings demonstrated that diminished SOCE in IP_3_Rs-TKO cells are caused by reduced expression of Orai1 protein, and that overexpression IP_3_R3 could largely restore both Orai1 levels and SOCE responses.

### 2.4. Overexpression of IP_3_R3 Restored SOCE by Inhibiting NEDD4L in GIPK Cells

To explore the underlying mechanism of reduced Orai1 protein level in IP_3_Rs-TKO cells, we first performed quantitative RT-PCR measurements. Results showed that mRNA levels of STIM1 and Orai1 in IP_3_Rs-TKO cells were similar to those in control cells (Figure 3D), indicating no changes at transcription level. We then examined whether Orai1 expression could be altered differently after blockage of protein synthesis with 50 μg/mL cycloheximide (CHX). Results showed that CHX did not restore the decrease of Orai1 protein levels in IP_3_Rs-TKO cells (Figure 3E and Appendix A), indicating no significant changes in the synthesis rate of Orai1. These results thus suggested that post-translation mechanisms might get involved in down-regulation of Orai1 protein levels.

Previous reports showed that Orai1 protein levels could be regulated by ubiquitination [45], a common post-translational modification that induces proteasomal degradation of proteins [46]. We therefore examined the effects of proteasome inhibitor MG132 (10 μM) on Orai1 protein levels. Unlike those in WT cells, 12-h treatments of MG132 significantly increased Orai1 protein levels in both types of IP_3_Rs-TKO cells (Figure 4A). These results suggested that IP_3_Rs-TKO could promote the degradation of Orai1 protein through the proteasomal pathway.

Lang’s works showed that Orai1 could be degraded via an ubiquitin protein ligase, NEDD4L [45,47]. We thus examined the effects of modulating NEDD4L activity on SOCE responses of GIPK cells. Compared to blank controls, GIPK cells expressing NL.1, a potent and specific intracellular activator of NEDD4L [48], had reduced SOCE. In contrast, the expression of a NEDD4L catalytic inhibitor NL.3 [48] had enhanced SOCE (Figure 4B,C). We then examined the effects of overexpression of NL.3 and NL.1 on Orai1 protein levels in GIPK cells. Western-blotting results showed that inhibition of NEDD4L with NL.3 significantly increased the protein levels of Orai1, while activation of NEDD4L with NL.1 further decreased Orai1 protein levels (Figure 4D). These results thus indicated that Orai1 levels and SOCE responses are modulated by NEDD4L-mediated ubiquitination. These results showed that the activity of NEDD4L E3 ligase could regulate Orai1 protein levels in GIPK cells.

To further understand the role of IP_3_R3 in the regulation of NEDD4L on Orai1 proteins and SOCE responses, we examined the effects of IP_3_R3 on mRNA levels of NEDD4L in GIPK cells using quantitative Real-time PCR. Compared to those in WT cells, the mRNA levels of NEDD4L were almost doubled in GIPK cells, suggesting possible presence of more NEDD4L protein in IP_3_Rs-TKO cells (Figure 4E), indicating more degradation of Orai1 via enhanced NEDD4L E3 ligase activity in GIPK cells. Moreover, overexpression of IP_3_R3 could partially decrease NEDD4L mRNA levels in GIPK cells (Figure 4E). This result indicated that IP_3_R3 could inhibit the activity of NEDD4L, restoring Orai1 protein levels and SOCE responses in GIPK cells (Figure 3B,C). The mechanistic underpinning of the linked expression between IP_3_R3 and NEDD4L is yet to be established. It is likely that some IP_3_R-dependent transcriptional factors might get involved and further investigations are needed to elucidate this.

## 3. Methods

### 3.1. Plasmids Construction

CDS sequences of mTurquoise and BFP were synthesized by Qinglan Biotech and Syngen tech, correspondingly. mNeonGreen plasmid and a plasmid containing mScarlet sequence were gifts from Dr. Chen Liangyi, Peking University. pBactSTneoB-mIP_3_R1/2/3 plasmids were gifts from Dr. Mikoshiba. To generate mScarlet-P2A-NL.1/NL.3, the sequence of mScarlet and NL.1/3 were amplified by PCR from WZ#325&649, WZ#329&653 [48], and mScarlet, correspondingly. The sequence of P2A was contained in the primer to connect to the C terminal of mScarlet. They were then inserted into pCDNA3.1(+) between the KpnI and EcoRV sites with the help of multiple fragment homologous recombination kit (Cat#: C113, Vazyme biotech, Nanjing, China). To generate mNeonGreen-P2A-mIP_3_R1, mTurquoise-P2A-mIP_3_R2 or BFP-P2A-mIP_3_R3, the corresponding sequences of fluorescent proteins and mIP_3_Rs were PCR-amplified, and then ligated into pcDNA3.1(+) with primers containing P2A sequence between NheI and NotI sites (Qinglan Biotech).

### 3.2. Cell Culture and Transfection

All the cells were grown in Dulbecco’s modified Eagle’s medium (DMEM, HyClone) containing 10% fetal bovine serum (Cleson Scientific), 1% penicillin and streptomycin (Thermo Scientific) and cells were maintained at 37 ℃ in a humidified atmosphere containing 5% CO_2_. Gene transfection was done with electroporation (Bio-Rad Gene Pulser Xcell system) using 4 mm cuvettes and OPTI-MEM medium. The electroporation protocol for HEK293 was a voltage step pulse (180V, 25 ms) [49]. 

### 3.3. Construction of Knockout Cell Lines

CRISPR/Cas9 technology was used to generate knockout cell lines. Where CRISPR stands for “Clustered regularly interspaced short palindromic repeats”, while Cas9 is the abbreviation for “CRISPR-associated protein 9”. All knockout cell lines containing ITPR1-ITPR2-ITPR3 triple KO (*itpr1/itpr2/itpr3*), ITPR2-ITPR3 double KO (*itpr2/itpr3*), ITPR1-ITPR3 double KO (*itpr1/itpr3*), ITPR1-ITPR2 double KO (*itpr1/itpr2*) were generated by CPISPR/Cas9 gene editing technology [17,50] from previously made GCaMP6m and R-CEPIA1er stable cells. sgRNA targeting sequences to these three genes were designed by SyngenTech company and inserted into lentiCRISPR v2 vector (Addgene plasmid no. 52961) by BsmBI site. The sequences of sgRNA set for three isoforms are shown in Table 1.

Protocols for the generation of KO cells were similar to those previously described [17]. Briefly, three sgRNA plasmids targeted ITPR1, ITPR2 and ITPR3 were transfected into cells by electroporation. Two days later, the transfected cells were selected with 2 μg/mL puromycin for 4 days. The survived multiple clonal cells were then seeded into 96 well dishes with a density of one cell/well. 10 days later, the resulting single clones were first functionally screened by calcium imaging and then the positive healthy clones were confirmed further with sequencing.

### 3.4. Single-Cell Intracellular Ca^2+^ Measurements

All Ca^2+^-imaging assay was performed in the HEK293 cells as described [17]. Briefly, intracellular Ca^2+^ imaging was performed with a ZEISS oberserver-A1 microscope equipped with a 40× oil objective (NA = 1.30), Lambda DG4 light source, and the SlideBook6.0 software (Intelligent Imaging Innovations, Inc.). GFP-1828A-000 filter and FURA2-C-000 filter were used to measure cytosolic Ca^2+^ signals indicated by GCaMP6m [29] and Fura-2 respectively, and Endoplasmic Reticulum (ER) Ca^2+^ levels indicated by R-CEPlA1er [28]. Fluorescence were obtained with a TxRed-A-Basic-000 filter set. The fluorescence signals from different Ca^2+^ indicators were collected every 2 seconds. The Ca^2+^ imaging solution contains 7.2 mM KCl, 107 mM NaCl, 1.2 mM MgCl_2_, 11.5 mM glucose, 20 mM HEPES-NaOH (pH 7.2).

Protocols for Fura-2 loading were similar to those described before [51]. Cells were first incubated with imaging solution containing 2 μM Fura-2 AM and 1 mM CaCl_2_ for 40 min at room temperature in the dark and subsequently eliminated Fura-2 AM for another 30 min. Data were acquired from collection of emission fluorescence at 509 nm generated by 340 nm excitation light (F340) and 380 nm light (F380) at 2 s intervals. The intracellular cytosolic Ca^2+^ levels were indicated by F_340_/F_380_ ratio.

### 3.5. Real-Time PCR

Total RNA was extracted from cells using TRIzol and the reverse transcription was performed with TransScript® First-Strand cDNA Synthesis SuperMix (Code#: AT301-03, TracsGen Biotech) following the manufacturer’s instructions later. Briefly, total RNA aliquots (3μg) were reverse transcribed at 65 ℃ for 5 min, ice-bath for 2 min, 42 ℃ for 30 min and 85 ℃ for 5 s. Reaction mixture aliquots (cDNA, 1 μL) were used as templates for two-step quantitative RT-PCR using TransStart® Top Green SuperMix (Code#: AQ131-03, TransGen Biotech). PCR running condition was 94 ℃ for 30 s, followed by 40 cycles of 94 °C for 5 s and 60 °C for 30 s. Then, the melting curve program was run. Relative differences in numerous mRNA expression levels was calculated by comparative cycle threshold method [52]. The primer targets for genes are listed in Table 2.

### 3.6. Cell Proliferation and Migration Essay

Cell proliferation essay were performed by IncuCyte^®^ Live Cell Analysis System (ESSEN BioSCIENCE), following the standard protocols provided by the company. For migration assay, cells were cultured into 96-well with a density of 30,000 cell/well, and EssenR 96-well WoundMaker^TM^ was used to make scratch wound after 5 h, and then the dish was put into IncuCyte that recorded the migration every 2 h for three days. The data were analyzed with IncuCyte^®^ ZOOM Software.

### 3.7. Western Blotting

Total proteins were extracted by Total Protein Extraction Kit (BB18011; BestBio) following the manufacturer’s instructions and concentration was measured by BCA protein assay kit (E162-01; GenStar). 6× loading buffer was added to protein extracts and boiled for 5 min at 60 ℃. 20 μg protein was separated on 8% SDS-PAGE and transferred onto PVDF membranes (Millipore). The resulting membranes were blocked for 1 h at 37 ℃ with TBST buffer (12 mM Tris-HCl, pH 7.5, 137 mM NaCl, 2.68 mM KCl, 0.1% Tween 20) containing 5% nonfat dried milk, then incubated with primary antibody overnight at 4℃. After washing three times (10 min each) in TBST buffer, the membranes were loaded with secondary antibody for 40 min. Detection was performed by ECL (GS009-4; Millipore) solution and imaged using Tanon5200 detection system finally. STIM1 and Orai1 was detected with anti-STIM1 antibody (5668S; CST) (1:1000 dilution) and anti-Orai1 antibody (o8264; Sigma) (1:300 dilution) followed by anti-rabbit-IgG (7074S; CST) (1:4000 dilution) respectively. Internal control GAPDH was detected with anti-GAPDH antibody (sc-32233; SANTA) (1:2000 dilution) and corresponding secondary antibody is anti-mouse-IgG (7076S; CST) (1:2000 dilution). The representative data shown were carried out from three independent experiments. The intensity of the images was quantified by Image J software, and the resulting data were plotted with prism7 software.

### 3.8. Proteasomal Inhibition Assays

Cells were seeded at 12-well plates, and were incubated for indicated hours respectively in DMEM high glucose with 10% FBS containing 50 μg/mL CHX or 10 μM MG132. We defined 0 h as 30 min-incubation. Incubated cells were washed in HBSS for second times and total proteins were extracted and immunoblotted as described above.

## 4. Conclusions

Overall, our results demonstrated that the expression level of IP_3_R3 correlated with SERCA pump activity and ER Ca^2+^ leakage rate, as well as SOCE amplitudes in HEK cells. These results indicate that IP_3_R3 is a crucial coordinator of ER-mediated Ca^2+^ signalling. Further studies are needed to understand the exact mechanisms underlying this coupling among different ER-Ca^2+^ signalling machineries in HEK cells.

## Figures and Tables

**Figure 1 cells-09-00275-f001:**
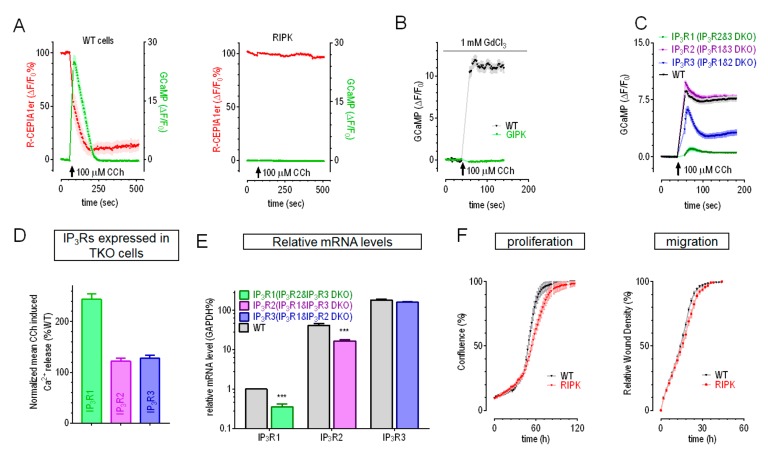
Characterization of HEK IP_3_Rs-TKO and IP_3_Rs-DKO cells. (**A**–**C**) Typical carbachol (CCh, 100 μM)-induced Ca^2+^ responses from ER in HEK IP_3_Rs-TKO and IP_3_Rs-DKO cells. (**A**) CCh-induced Ca^2+^ changes as shown with transiently expressed cytoplasmic Ca^2+^ indicator GCaMP6m (Green) or stably expressed ER Ca^2+^ indicator R-CEPIA1er (Red). Left panel, responses of WT HEK cell stabling expressing R-CEPIA1er; right panel, R-CEPIA1er stable cells with IP_3_Rs-TKO (RIPK). (**B**) Representative CCh-induced responses in HEK GCaMP6m stable cells without (WT, black trace) or with IP_3_Rs-TKO (GIPK, green trace). (**C**) Typical CCh-induced Ca^2+^ releases in HEK GCaMP6m WT cells or those with IP_3_Rs-DKO. 1 mM GdCl_3_ was included in bath solution to block Ca^2+^ movements across PM in B) and C) (n = 3). (**D**) Statistics showing relative sizes of mean CCh-induced Ca^2+^ releases in RIPK or GIPK cells transiently expressing IP_3_Rs (see Appendix A for typical curves). (**E**) Relative mRNA levels of three types of IP_3_Rs in HEK cells and IP_3_Rs-DKO. mRNA levels were first normalized against corresponding GAPDH levels, then normalized against corresponding IP_3_R1 levels of WT cells. Expression levels of IP_3_R1 in WT cells were set as 1 (mean ± SEM, *** *P* < 0.0001, Student’s *t*-test). (**F**) Typical proliferation (left panel) and migration (right panel) curves of RIPK and WT cells.

**Figure 2 cells-09-00275-f002:**
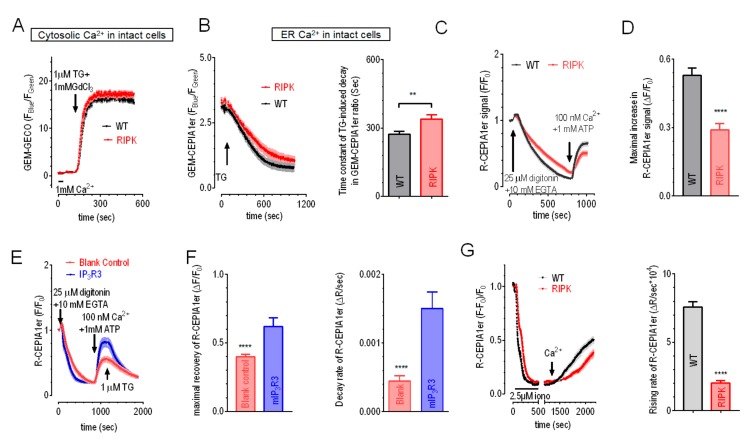
Effects of IP_3_Rs-TKO on ER Ca^2+^ dynamics and corresponding changes induced by overexpression of mIP_3_R3. (**A**) Typical traces showing TG (1 μM)-induced Ca^2+^ releases in HEK WT (black trace) or RIPK (red trace) cells transiently expressing GEM-GECO. 1 mM GdCl_3_ were included in external solution to block Ca^2+^ movement across PM. No differences in responses between WT and RIPK cells were seen (n = 2). (**B**) TG (1 μM)-induced responses indicated with transiently expressed GEM-CEPIA1er. Left, typical traces; right, statistics (n = 3, ** *P* < 0.01, Student’s *t*-test). (**C**) Typical changes in R-CEPIA1er signals in response to PM permeabilization (25 μM digitonin), passive ER store depletion (10 mM EGTA), and SERCA-mediated ER Ca^2+^ refilling in the presence of ATP and Ca^2+^. (**D**) Statistics of traces in C) showing maximal increases in R-CEPIA1er signals after the addition of Ca^2+^ and ATP (n = 4, *** *P* < 0.0001, Student’s *t*-test). (**E**) Representative R-CEPIA1er traces of RIPK cells transfected either with empty vector or BFP-P2A-mIP_3_R3. Experimental procedures were similar to those in (**C**) in the first steps. In the end, 1 μM TG was added into bath solution to examine ER Ca^2+^ leakage. (**F**) Statistics of (**E**). Left, maximal increases in R-CEPIA1er signals after addition of ATP and Ca^2+^; right, initial decay rate of R-CEPIA1er signals after TG. (**G**) ER Ca^2+^ refilling indicated by R-CEPIA1er signals in intact R-CEPIA1er or RIPK cells. ER Ca^2+^ stores were depleted with 8-min bath with 2.5μM ionomycin. After another 10-min in 0Ca^2+^ solution, Ca^2+^ were added externally to allow SOCE and corresponding ER Ca^2+^ refilling. 300μM Ca^2+^ was used here to enlarge possible differences in SOCE amplitudes between two types of cells. Left, typical traces; right, statistics (n = 3, *** *P* < 0.0001, Student’s *t*-test).

**Figure 3 cells-09-00275-f003:**
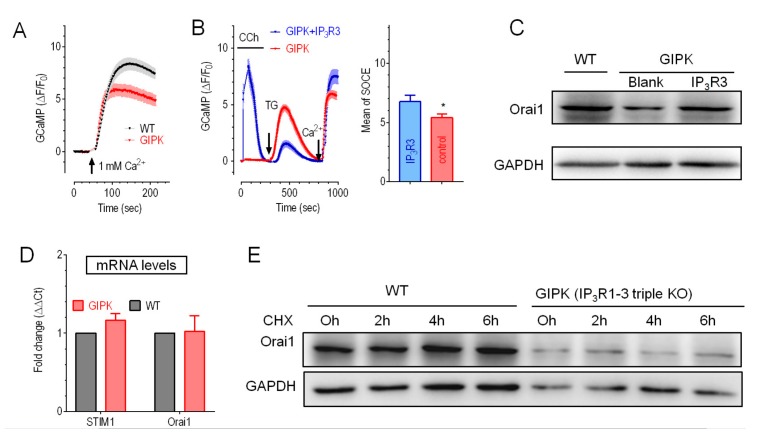
Characterization of SOCE and Orai1 protein levels in IP_3_Rs-TKO cells. (**A**) Typical TG-induced SOCE responses of HEK GIPK or WT GCaMP cells measured with cytoplasmic Ca^2+^ indicator GCaMP6m. Before recordings, ER stores were emptied by 10-min-pretreatement with 1 μM TG. Compared to corresponding WT cells, mean SOCE in GIPK cells were significantly reduced (7.4 ± 0.3 Vs 5.6 ± 0.3, n = 3, *p* < 0.0005, Student T-test). (**B**) CCh (100 μM)-and TG (1 μM)–induced Ca^2+^ responses in GIPK cells transiently expressing empty vector or BFP-P2A-mIP_3_R3. Left, typical traces; right, statistics of SOCE responses, (n = 3, p < 0.05, Student T-test). (**C**) Representative Western blotting results showing the effects of IP_3_R3-overexpression on Orai1 protein levels in GIPK cells. Densitometric analysis of all repeats showed that the protein levels of STIM1 were unaltered (*P* > 0.5, Student’s *t*-test, n = 3), while those of Orai1 were significantly reduced (*P* < 0.05, Student’s *t*-test, n = 3). (**D**) Relative mRNA levels of STIM1 and Orai1 in GIPK cells performed by quantitative RT-PCR. Relative mRNA expression levels of target genes were first normalized against those of GAPDH, then normalized against corresponding WT cells. (*P* > 0.5, Student’s *t*-test, n = 3). (**E**) Representative Western blotting results. Densitometric analysis of all repeats showed that Orai1 protein level did not significantly change in WT GCaMP or IP_3_Rs-TKO (GIPK) cells after treatments with CHX (50 μg/ml), a known translation inhibitor, (*P* > 0.2, Student’s *t*-test, n = 3).

**Figure 4 cells-09-00275-f004:**
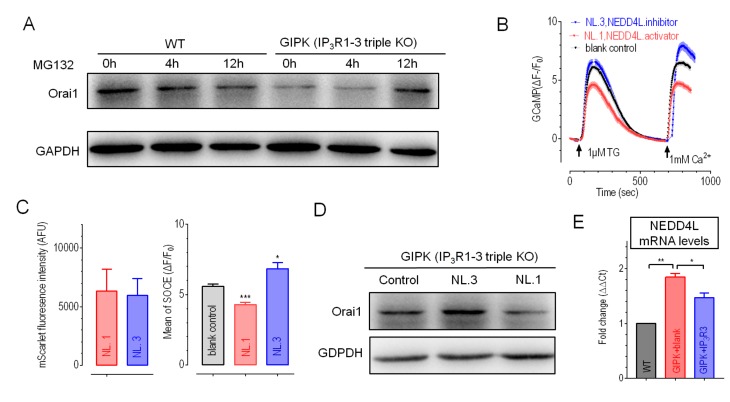
The expression of Orai1 and SOCE amplitudes in IP_3_Rs-TKO cells could be regulated by modulating activities of NEDD4L. (**A**). Representative Western blotting results of GIPK or WT GCaMP cells after incubation of MG132 (10 μM), a proteasome inhibitor. Densitometric analysis of all repeats showed that, after 12-h treatments, Orai1 levels in GIPK cells were significantly increased. (Normalized against GAPDH, * *P* < 0.05, Paired *t*-test, n = 3). (**B**–**C**). Effects of NL.1 or NL.3 overexpression in GIPK cells. (**B**), Typical traces; (**C**), Statistics. Left, statistics of mScarlet fluorescence, indicating the relative expression of mScarlet-P2A-NL.1/NL.3; Right, statistics of mean SOCE responses shown in (**B**). Compared to blank controls, inhibition of NEDD4L with NL.3 significantly increased SOCE, while NL.1 significantly reduced SOCE in GIPK cells (* *P* < 0.03; *** *P* < 0.002, Student’s *t*-test, n = 3). (**D**) Typical Western blotting results. Densitometric analysis of all repeats showed that both NL.1 and NL.3 significantly altered Orai1 protein levels (NL.1 vs blank, *P* < 0.02; NL.3 vs blank, *P* < 0.04; Student’s *t*-test, n = 3). (**E**) mRNA levels of NEDD4L in WT or GIPK cells with or without overexpression IP_3_R3. (** *P* < 0.002; * *P* < 0.02, Paired *t*-test, n = 3).

**Table 1 cells-09-00275-t001:** Sequences of sgRNAs and sequencing primers used for knocking out IP_3_Rs in HEK293 cells.

Target	sgRNA Sequence (5′→3′)	Target CDs Site	Primers (5′→3′)
ITPR1	CAAAGACGACATATTAAAGG	687–706	F: TGCTGTGATTTTAGTGGCGT R: TCTCCACCCTACCCTTACCT
ITPR2	ACACGATGTCCCCTATGTAG	46–27	F: TCTTGGCCGCTGTAGTCC R: CCAGGGAAAACAAGCACACA
ITPR3	ACCCCCCTTCTCACGGAACG	1593–1574	F: GGGTTGTGGTCCAGCTTAGA R: CATCTAACCCAGTGCAAGGC

**Table 2 cells-09-00275-t002:** Primer sequences for quantitative real-time PCR.

Gene	Forward Primer	Reverse Primer	NCBI Reference Sequence
ITPR1	ATGACAGCTCTGAGGAGAA	GCAGAAGAGACAGGAGATTTAG	NM_001099952.2
ITPR2	CTGTGTGGAAGAGCAACTAC	ATGGGTCTGTAGGTAGGAATAG	NM_002223.3
ITPR3	CAAGCCCTCCAAAGATGAG	CGAAGGTGTCGATGATTACC	NM_002224.3
STIM1	GGATCTCAGAGGTTGGTAGA	GATGGAAGAGGAGCAAGAAG	NM_001277961.1
Orai1	GACGCTGACCACGACTA	CCTTGACCGAGTTGAGATTG	NM_032790.3
GAPDH	AACTGCTTAGCACCCCTGGC	ATGACCTTGCCCACAGCCTT	NM_001256799.2
NEDD4L	CCTAGATTGTCACGGTTTCC	GACTAAACTCTCACCTCCTTTC	NM_001144964.1

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
