# Peer review of "Type 3 Inositol 1,4,5-Trisphosphate Receptor is a Crucial Regulator of Calcium Dynamics Mediated by Endoplasmic Reticulum in HEK Cells"

_cells, 2020, doi:10.3390/cells9020275_

Round 1

Reviewer 1 Report

Yue et al describes in this paper the consequences of IP3R deletion in ER calcium dynamics and homeostasis. The major finding is that there is an impaired ER Ca2+ leak and refilling in IP3R-TKO cells that might be linked to the increased expression of the ubiquitin ligase NEDD4L that promotes the degradation of Orai1 protein through proteasome. Unfortunately, the paper is not very interesting for major reasons:

1- All the observations are made in HEK293 cells genome-edited to deplete IP3Rs. Similar observations should be made in more relevant cell lines where depletion of IP3Rs have been revealed important (see any review in the references of the current manuscript)

2- How many clones have been analyzed. In general CRISPR/Cas9 cell lines should be studied as single clones and several clones should be tested (at least for the most important observations) in order to confirm the results.

3- The data about the role of NEDD4L in Ora1 degradation is probably the most interesting piece of data. However, the analysis of NEDD4L is very superficial. Control of NL.1 and NL.3 overexpression is not presented in the IBs and Loss of function as well as gain of function experiments should be performed to further link the expression of NEDD4L to Ora1 degradation.

4- It is not clear how the depletion of IP3Rs can induce the transcription of NEDD4L. There are several transcription factors that are altered upon ER stress or IP3R deletion (i.e. NFAT). The authors should try to at least discuss a potential hypothesis. Better if they perform any experiment regarding this concern.

Reviewer 2 Report

The paper by Wang and co-workers is an interesting paper which aims to dissect the contribute of IP3 receptors and of its three different isoforms to the Ca2+ handling of the intracellular stores. The experiments are well designed, and the results accurately discussed, however I believe that the conclusions that the authors drew need to be better discussed and supported by additional control experiments.

Specifically, I have several major concerns:

In general, all the experiments and the conclusions were carried out by analysing one single clone of IP3 TKO cells for IP3R (GIPK and RIPK) and as the authors admitted on page 6 line 243 for DKO cells a long term adaptation can be occurred. Different clones for each type should be considered in Ca2+ analysis and some parameters which do not required the expression of the specific cytosolic or ER Ca2+ indicators, e.g., cell survival or proliferation and migration, should be verified for both GIPK and RIPK clones to independently assess the effect IP3R KO and reduce the possibility that the effects could be dependent on clonal selection The effect of IP3 R downregulation on NEDD4 L mRNA level is totally unexplained. It is necessary to explore it better but first of all if it is specifically linked to the absence of IP3 Rs or rather is a clonal peculiarity. In any case a possible explanation for this finding need to be discussed- What is the author hypothesis? It is necessary to check the level of NEDD4 protein by western blotting. And again, if this is the case it is unlikely that the increase of NEDD4 protein activity specifically affected Ora1 levels. The level of other substrate proteins for NEDDilation should be verified as control to demonstrate the link with increased activity.

 In details:

Page 3 lines 110-115 and figure 1D the authors should show and compare the level of IP3R overexpression in respect with the endogenous content in not silenced cells. The result could be affected by the fact that upon overexpression the amount of IP3 receptor is much more that the endogenous content and this could make a difference in the interpretation of the results.

Pages 3-4 lines 126-131: first of all, the proliferation and migration rates (fig. 1F) should be evaluated in different clones of RIPKs as well as in GIPK cells. They all should behave at the same way since the two cell clones only differ for the type of the expressed Ca2+ indicator.

Page 3 line 121 “quantitative RT-PCR results showed that mRNA levels of IP3R2 and IP3R3 is far higher than that of IP3R1 in WT 122 cells”. And what about the protein levels?

The author should perform a western blotting.

Page 4 line 129 “the proliferation rate of RIPK was significantly reduced …” . I would not say that the proliferation is reduced but rather delayed since at 120 h the confluence is the same as in the control cells. And again, until about 50 h the rate of proliferation is the same between wt and RIPK. The significance is not reported. Migration rate seems to be unaffected rather that slightly inhibited. Please comment these results appropriately.

Page 4 “RIPK cells adapt well from IP3-TKO….” Please comment these findings in respect to the paper by Foskett and co-workers Cell. 2010 Jul 23;142(2):270-83. doi: 10.1016/j.cell.2010.06.007

Page 5 line 169 and Figure 2B. ER Ca2+ levels depend by the balance between Ca2+ leak and SERCA pump activity as the authors stated, but it should take into account that the action of calcium buffering proteins, such as calreticulin, is determinant in regulating the amount of free Ca2+ (not buffered)  in the ER lumen and that Ca2+ buffering is tunable. The authors should consider this and comment appropriately.

Page 5 line 179  and Figure 2C. According to these experiments RIPK cells have slower rate and slower amplitude of Ca2+ replenishment in the ER lumen upon store depletion, suggesting that basal ER Ca2+ level could be reduced. But this result is in contrast with that shown in Figure 2B indicating that the level of ER Ca2+ is not affected by IP3R KO. The authors should comment these discrepancies. Does it mean that in intact vs permeabilised cells some compensation due to increase capacitative Ca2+ influx could occur? However, this seems not to be the case since SOCE is smaller in RIPK cells as documented by Figure 2D and 2G and Figure 3A. And also ORai levels is reduced in IP3R TKO cells

The western blottings shown in Figure 3 and 4 together with similar replicates need to be quantified by densitometric analysis to establish possible difference in the level of expression. The bands relative to WT in the WB of Figure 3E are saturated, a blot with a reduced exposure should be used for the quantification.

Page 7 line 254. Fig 2C is not correctly quoted. Here it is Figure 3C

Reviewer 3 Report

The study entitled 'Type 3 inositol 1, 4, 5-trisphosphate receptor is a crucial regulator of calcium dynamics mediated by endoplasmic reticulum in HEK cells' is an original paper exploring the role played by the IPR3 receptor in ER-mediated calcium signaling. The authors did a great effort and used both double and triple knockout HEK cells lines in order to examine calcium dynamics and CRAC channel activity in these cells. 

The authors identifyed that IP3R2 is the main isoform that contributes to the IP3-induced calcium release in endoplasmic reticulum. Additionally, the authors demonstrated that the IP3R1 and IP3R3 isoforms are essential players in the refilling / leakage mechanisms involved in ER calcium dynamics, and that IP3R3 isoform balanced Orai1 / SOCE expression.

I have several suggestion for the authors to improve their manuscript

Please add a separate subsection of Conclusions. Several abbrevations used in the text should be explained. 

Reviewer 4 Report

This manuscript reports roles for subtypes of IP3R in the HEK cell IP3R triple KO cell line. The authors demonstrate that ER calcium leakage, SERCA pump-dependent calcium reuptake and SOCE are all partially controlled by the presence of IP3Rs. IP3R3 subtype plays a dominant role in controlling Orai1 protein levels and SOCE through NEDD4L-dependent degradation of the channel protein. The experiments seem well carried out and the findings are clear. The paper needs language editing in places as the text is occasionally ahrd to follow. What the paper also lacks is a mechanistic explanations of the observations. How does the absence of IP3Rs slow down SERCA activity? And how does the IP3 R3 regulate NEDD4L activity? These points should be discussed. The authors show that over expression of IP3Rs can rescue ER calcium homeostasis but does this require calcium release activity? Is SOCE rescued for example when a pore-dead IP3R is put back instead? These are simple experiments to do but would provide valuable insight into mechanism.

Round 2

Reviewer 2 Report

Authors' response: "Please refer to our response to Comment 2 by Reviewer 1 regarding the number of clones analyzed"

Unfortunately, I have no access to the other reviewer's comments and response. Could you attached your response on this point also to me?

I am not fully agree on this aspect. Several , I believe indicate that calretinin could have an important role also in ER Ca free  levels in addition to Ca2+ release and SOCE. In any case since both Ca2+ release and SOCE are investigated in the present paper I believe that the role of calreticulin or ER Ca2+ buffering by other proteins should be better discussed in the light of the present literature.

Here some related papers just to mention a few of them:

1: Llewelyn Roderick H, Llewellyn DH, Campbell AK, Kendall JM. Role of
calreticulin in regulating intracellular Ca2+ storage and capacitative Ca2+ entry
in HeLa cells. Cell Calcium. 1998 Oct;24(4):253-62. PubMed PMID: 9883279.

2: Xu W, Longo FJ, Wintermantel MR, Jiang X, Clark RA, DeLisle S. Calreticulin
modulates capacitative Ca2+ influx by controlling the extent of inositol
1,4,5-trisphosphate-induced Ca2+ store depletion. J Biol Chem. 2000 Nov
24;275(47):36676-82. PubMed PMID: 10973951.

3: Bastianutto C, Clementi E, Codazzi F, Podini P, De Giorgi F, Rizzuto R,
Meldolesi J, Pozzan T. Overexpression of calreticulin increases the Ca2+ capacity
of rapidly exchanging Ca2+ stores and reveals aspects of their lumenal
microenvironment and function. J Cell Biol. 1995 Aug;130(4):847-55. PubMed PMID:
7642702; PubMed Central PMCID: PMC2199966.

Reviewer 4 Report

The authors have addressed my comments by modifying the discussion a little. It is a pity the authors were reluctant to carry out some straightforward experiments to provide mechanistic insight into their findings, which remain purely descriptive. 
